# Identifying the impact of social influences in health-related discrete choice experiments

**Esther W. de Bekker-Grob**[1,2]*, **Kirsten Howard**[3], **Joffre Swait**[1,2]

**1** Erasmus Choice Modelling Centre, Erasmus University Rotterdam, Rotterdam, The Netherlands, **2** Erasmus School of Health Policy & Management, Erasmus University Rotterdam, Rotterdam, The Netherlands, **3** School of Public Health, Faculty of Medicine and Health, University of Sydney, Sydney, Australia

* debekker-grob@eshpm.eur.nl

## Abstract

Several disciplines, among them health, sociology, and economics, provide strong evidence that social context is important to individual choices. It is therefore surprising that relatively little research has been focused on integrating the effect of social influence into choice models, especially given the importance of such choices in healthcare. This study developed and empirically tested a choice model that accounts for social network influences in a discrete choice experiment (DCE). We focused on maternal choices for childhood vaccination in Australia, and used an econometric choice model that explicitly 1) incorporated vaccine schedule characteristics, benefits and costs, and 2) represented up to ten different identifiable key influencer types (e.g., partner, parents, friends, healthcare professionals, *inter alia*), allowing for the attribution of directional importance of each influencer on the gravid woman's decision to adhere to or reject childhood vaccination. Pregnant women (N = 604) aged 18 years and older recruited from an online panel completed a survey, including a DCE and questions about key influencers. A two-class ordered latent class model was conducted to analyse the DCE data, which assumes that the underlying latent driver (in our case the WHO vaccine hesitancy scale) is ordered, to give a practical interpretation of the meaning of the classes. When the choice model considered both childhood vaccination attributes and key influencers, a very high model fit was reached. The impact of key influencers on maternal choice for childhood vaccination was massive compared to the impact of childhood vaccination attributes. The marginal impact differed between key influencers. Our DCE study showed that the maternal decision for childhood vaccination was essentially almost completely socially driven, suggesting that the potential impact of social network influences can and should be considered in health-related DCEs, particular those where there are likely to be strong underlying social norms dictating decision maker behaviour.

## 1. Introduction

Choice modelling is mainstream in marketing, transport and environmental economics, where it is used–among other applications–for transport policy development, urban planning

**Data Availability Statement:** The data underlying the results presented in the paper and the corresponding README_file are publicly available

via Dryad (see https://doi.org/10.5061/dryad.fqz612jvr).

**Funding:** Grant support was from Australian Research Council (https://www.arc.gov.au/) Discovery Project DP140103966 to JS, and from The Netherlands Organisation for Scientific Research (https://www.nwo.nl/en) (NWO-Talent-Scheme-Vidi-Grant No. 09150171910002) to EBG. The funders had no role in the study design, the collection, analysis, decision to publish, or preparation of the manuscript.

**Competing interests:** The authors have declared that no competing interests exist.

systems, marketing product pricing, and resource management decision-making [1, 2]. Choice modelling is also increasingly used in health economics. Since its introduction in the early nineties, the stated choice technique 'Discrete Choice Experiment (DCE)', originating from mathematical psychology [3], has become increasingly popular [4–7]. It has been used–among other applications–for informing resource planning and predicting uptake where there is no information or trial data [4–7]. Although choice modelling via DCEs has a huge potential to support decision-making in healthcare, some of the usual assumptions made in their application limit their usefulness. Traditionally, almost universally in health, transport, marketing and environment applications of choice models assume that choice processes are independent of the influence of other people than the decision-maker (e.g., the patient, the physician, health policy maker) in question (or rather, they simply do not explore understanding of preferences). However, and most especially in healthcare, choices are not made in a social vacuum.

It is surprising that relatively little research has been focused on integrating the effect of social influence into choice models [8], especially given the importance of such choices in healthcare [9, 10]. This statement is made with the recognition that several disciplines, among them health, sociology, and economics, provide strong evidence that social context is important to individual choices (see, e.g., the health literature concerning MMR vaccine to protect against measles, mumps, and rubella [11–13]). At most, outside the healthcare area, there is existence of econometric corrections that might arise from supposed social influences [14], and approaches to measure and model joint choices (e.g. household decisions) or effects of attitudes of peers on the decision-making process of the individual [8, 15].

As most DCE surveys in healthcare are not designed to isolate social influences arising from the decision-makers' context, the aim of this study was to develop and empirically test a choice model that accounts for social network influences in a health-related DCE. Hence, in our study, we seek to isolate this one aspect of context, which we thought a priori should have a substantial impact on decision-makers: influences of social network. That is, careful thought suggested to us that this particular source is a potentially very important type of influence establishing people's behaviour. To test this hypothesis, the choice model we developed—as preferences are one way to understand influences of social network on decision-makers—was expanding the decision model for this particular type of influence.

We focused on maternal choices for childhood vaccination in Australia, and used an econometric choice model that explicitly 1) incorporated vaccine schedule characteristics, benefits and costs (i.e. vaccine schedule attributes), and 2) represented up to ten different identifiable key influencer types (including partner, parents, friends, healthcare professionals), allowing for the attribution of importance (and direction) of each influencer on the gravid woman's decision to adhere to or reject childhood vaccination. As the DCE method is a richer method compared to the direct survey question–that is, it is able to jointly take childhood vaccination schedule attributes and social influences into account, while also considering the respondent's trade-offs between the variables, between key influencers and between variables and key influencers–we hypothesise that the DCE outcomes are more informative and therefore more useful for policy decision-making (e.g., identification and targeting of particular messages or conveyers of messages to particular respondent types).

## 2. Methods

For several reasons we focused on childhood vaccination decisions (more specifically, vaccinations over the first 12 months) in Australia: (a) subjects' choices for childhood vaccination are likely highly sensitive to social influences due to the strong underlying social norm (including

current Australian policy response [16] to vaccine hesitancy [17]), which is vital to empirically test a choice model that accounts for social network influences; (b) declines in childhood vaccination rates in Australia (and worldwide) have received media coverage in the last few years, contributing to the relevance of the study; and (c) understanding broader influences on vaccination decisions may help determine targeting of appropriate education, behaviour change strategies and/or policy. A labelled DCE survey [18], that was suitable to explain real-life choices and able to deal with social influences, was conducted. Approval for the study was obtained from the Human Research Ethics Committee, University of South-Australia (PG082203).

## 2.1. Literature review and attribute selection

Childhood vaccination decisions can be conceptualised as occurring along a continuum, with full adherence to recommended vaccination schedules at one end and no vaccination at the other. That is, childhood vaccination is a preventive intervention, so individuals are not obliged to get vaccinated against diseases. Therefore, parents may choose to delay or split scheduled vaccinations, or to partly vaccinate. This so-called vaccine hesitancy [17] is however not without risk. It has significant effects on population disease risk [19, 20]. To identify vaccine schedule attributes (including attribute levels) that may impact maternal choices to opt for 1) the recommended vaccination schedule, 2) a delay or split scheduled vaccination, or 3) no vaccination at all, we used a literature search among stated preference and qualitative studies in childhood vaccination [21–34], and the Australian National Immunisation Program (NIP) Schedule for young children as at August 2015. Thereafter, based on the literature review and the Australian Immunisation Schedule, the most relevant childhood vaccination schedule attributes and their consequences were included in the labelled DCE. The attributes and their alternative specific levels are presented in Table 1.

## 2.2. Discrete choice experiment (DCE)

The labelled DCE design contained three alternatives for each choice task (i.e., a recommended vaccination schedule profile (fixed), a certain delay or split scheduled vaccination profile (variation), and a 'no vaccination' profile (fixed)). We deliberately chose to mirror the real choices that parents would be making with respect to vaccination–a fixed schedule, as per recommended (A); a variable schedule with possible delays/splits (B), and a fixed no vaccination schedule (C)–with only schedule B varying in each question. In each choice task, respondents were asked to opt for the alternative that appealed most to them. The combination of eight attributes and their corresponding levels resulted into many potential delays/splits scheduled childhood vaccination profiles. To create a much smaller subset of childhood vaccination scenarios (i.e. choice tasks) with little loss of information or estimation precision, a Bayesian D-efficient DCE design was used [35]. We generated a DCE design consisting of 600 choice tasks blocked into 50 sub-designs using NGene software [36]. Each respondent was randomly assigned to a sub-design containing 12 discrete choice tasks each.

To reduce respondent burden further and to be as clear as possible, the attribute levels of 'the delay or split scheduled vaccination' profile were presented as a direct comparison with the 'fixed recommended vaccination schedule' profile. See S1 Appendix for more details. The information we presented about the attributes and their levels, was consistent with the Australian NIP Schedule as at August 2015, existing and newly considered government policies with respect to consequences for delayed or no vaccination and parental information brochures provided by the NIP and general practitioners to minimise information asymmetry between the hypothetical situations representing the actual decision and the actual decision. Hence,

**Table 1. Attributes and alternative specific levels for childhood vaccination in Australia.**

| Attributes | Alternative Specific Levels |
|---|---|
| Number of vaccines that are delayed | |
| Recommended vaccination schedule | 0 |
| Delay/split some vaccs or visits | 0-1-2-3 |
| No vaccinations | 0 |
| Number of vaccines that are split | |
| Recommended vaccination schedule | 0 |
| Delay/split some vaccs or visits | 0-1-2-3-4 |
| No vaccinations | 0 |
| Number of injections over 12 months | |
| Recommended vaccination schedule | 9 |
| Delay/split some vaccs or visits | 1-2-3-. . .-19 |
| No vaccinations | 0 |
| Number of visits to the clinic for vaccinations over 12 months | |
| Recommended vaccination schedule | 4 |
| Delay/split some vaccs or visits | 1-2-3-. . .-16 |
| No vaccinations | 0 |
| Chance the child will get a vaccine preventable disease (per 1000) | |
| Recommended vaccination schedule | 1 |
| Delay/split some vaccs or visits | 2-3-. . .-11 |
| No vaccinations | 12 |
| Chance the child might experience minor side effects (per 100) | |
| Recommended vaccination schedule | 20 |
| Delay/split some vaccs or visits | 20-30-50 |
| No vaccinations | 0 |
| Implications for formal childcare arrangements as a result of vaccine status | |
| Recommended vaccination schedule | Child can enrol in childcare as normal |
| Delay/split some vaccs or visits | Child can enrol in childcare as normal—Child will be excluded from childcare if there is an outbreak |
| No vaccinations | Child cannot enrol in childcare at all |
| Implications for government family assistance payments as a result of vaccine status | |
| Recommended vaccination schedule | Parent is eligible for all normal family assistance payments |
| Delay/split some vaccs or visits | Parent is eligible for all normal family assistance payments - |
| | Family assistance payments for the child will be reduced |
| No vaccinations | Parent will not receive family assistance payments for the child |

attribute wording was deliberately made consistent with the words and descriptions that were used in the NIP Schedule and are therefore the words that parents would be presented with when making the actual decisions about vaccination. Informal piloting of these attribute descriptions did not indicate any difficulties with understanding and interpretation of the wording.

## 2.3. Survey and sample

Besides the 12 DCE choice tasks (for the estimation of the decision model) and questions regarding respondents' characteristics, the survey included questions where respondents were asked to consider and assess statements about vaccines and vaccinations using validated questions from Zingg et al. [37] and Larson et al. [38]. That is, to be able to detect substantial correlations between knowledge and willingness to vaccinate, Zingg et al. [37] developed a one-dimensional knowledge scale about vaccination with good psychometric properties. To help diagnose and address vaccine hesitancy, Larson et al. [38] developed a matrix that mapped the key factors influencing the decision to accept, delay or reject some or all vaccines under three categories: contextual, individual and group, and vaccine-specific. Additionally, respondents were asked to assess their susceptibility to social influences with respect to health behaviours using validated questions from Holt et al. [39]. That is, to answer some of the questions about the role of social networks and social influence processes on health behaviours and outcomes, Holt et al. [39] developed and validated an instrument to assess the perceived role of others in the health behaviour decisions of individuals. Respondents were also asked about different people ('key influencers' or KIs) in their life and the role KIs might play in their decision to vaccinate their child. This included questions such as 'Would this person(s) be a possible source of health information?'; 'How long have you known this person?'; 'How important is this person's view on your decision to vaccinate your child?'; and 'For <key influencer type X>, please tell us how likely you believe they would be to recommend each vaccination option by allocating 100 points across the three options for this person, with more points meaning that they are more likely to recommend that option'. The survey also included questions related to childhood vaccination knowledge, the WHO vaccine hesitancy scale (which mapped key factors influencing the decision to accept, delay or reject some or all vaccines under three categories: contextual, individual and group, and vaccine specific),[38] and attitudinal questions. The DCE survey is available from the authors upon request.

An online sample of 604 pregnant women from the Australian general population, nationally representative in terms of age, education, and geographic region was recruited via Survey Sampling International (currently known as Dynata) and Bounty (an Australian parent specific panel; www.bountyparents.com.au/). Calculation of optimal sample sizes for a DCE is complicated as it depends on the true values of the unknown parameters in the assumed discrete choice model [40]. Taking the aim of our study into account, we opted to be parsimonious and focus on identifying the relative role of social influences of maternal choices (hence, we were less interested in how accurate the model could be with respect to uptake estimation of a certain childhood vaccination program – i.e., we were not setting out our research to predict vaccine uptake as accurately as possible). Accordingly, our sample size was based on affordability, which suggested using a convenience sample with a sample size that was sufficient for study purposes. See S2 Appendix for more details about the sample size calculation. The respondents were incentivized according to the online panel companies' incentive scheme. Written informed consent was obtained from all participants for inclusion in the study. We utilized all respondents that completed (i.e., did not abandon) the survey before it was closed off at the end of the data collection period (see S3 Appendix for more details).

## 2.4. DCE analysis

Several models exist to analyse discrete choice data [7, 41, 42]. Each choice model has its set of features, which should fit the intentions of the research and match the respondents' choice behaviour. Given our interest in accounting for social network influences, while also taking our sample size and the risk of overfitting into account, led to the decision to employ (after exploring a traditional multinomial model logit (MNL) model first) a panel ordered latent

class model (OLCM) using custom-developed software in Fortran 77. A latent class model can be used to identify the existence and the number of segments or classes, $M$, in the population (i.e., identifying different utility (preference) functions across unobserved subgroups). Class membership is latent in that each respondent belongs to each class up to a modelled probability and not deterministically assigned by the analyst a priori. It is noteworthy that the ordered LCM model is flexible in that the probability that sampled respondents belong to a particular class can be linked to the underlying ordinal latent driver (in our case the WHO vaccine hesitancy scale), hence allowing for some understanding as to the make-up of the various class segments (i.e., a neat way to classify respondents). (See more details in the S4 Appendix.) This additional information can be really useful for policy makers to help them to develop appropriate and tailored education, behaviour change strategies and/or policy. We used a *panel* LCM, as it accounts for the panel nature of the data since each respondent completed 12 discrete choice tasks. We systematically tested standard (i.e. nominal) and ordered LCMs with different classes, and several different specifications for the utility function (e.g., categorical or numerical attribute levels, two-way interactions between attributes, several attribute transformations). To account for social network influences and modelling our DCE data by assuming a utility maximisation decision process, we used the conditional choice model formulated by Swait & Marley [43] extended for multiple influencers (see S4 Appendix for more details).

The final model was based on parsimoniousness and Bayesian Information Criterion outcomes and resulted in a two-class ordered latent class (2-OLCM) model, which assumed that the underlying classes are ordered on a single latent dimension (in our case, the WHO vaccine hesitance scale) [44]. Ordered latent class models are not new to the econometric literature (e.g., Gopinath and Ben-Akiva [45]; Swait and Sweeney [46]; Swait and Adamowicz [47]), but its frequency of use is much lower than its categorical counterpart (e.g., among many others, Huls et al. [48]). We detail the full model in the S4 Appendix, to which we direct the reader, whereas in S5 Appendix more details about the DCE analysis can be found.

When the choice model considered both childhood vaccination attributes and key influencers, a very high model fit was reached. The utilities of the different vaccination schedules are specified on the basis of two types of factors: (1) schedule characteristics (see Table 1) and (2) key influencer impacts. To interpret the schedule characteristics coefficients, a positive effect indicates that schedule characteristic makes vaccination more likely, while a negative effect indicates vaccination becomes less likely with increases.

The key influencers are represented with terms such as 'ln(Partner), ln(Mother), . . .', which describe the potential effect on the respondent's choice of that actor through the respondent's perception of what would be the actor's recommended behaviour. These key influencer effects are further deemed to be potentially moderated by two respondent characteristics: their level of knowledge about vaccination (K) and their susceptibility to social influences (S). These moderating characteristics were measured using established scales (for K, see Zingg and Siegrist [37]; for S, see Holt et al. [39]).

The sign of the key influencer coefficients (including moderation interactions) reflects whether the key influencer has a positive or negative effect on the utility for childhood vaccination schedule. The magnitude of the class assignment threshold $\tau$ indicates the cut-off value of the WHO vaccine hesitancy scale separating two (consecutive) classes. That is, the probability to belong to a specific class is determined by the WHO vaccine hesitancy scale value.

## 2.5. Impact of social influences on childhood vaccination decision

Using the estimated coefficients from the ordered 2-class model, the average marginal impact of each key influencer (KI) on the utility of a vaccination schedule was determined. By

assumption/construction of the choice models, the estimated effects for each KI are interpreted as orthogonal (or independent) of the others; this assumption is held in common with all regression-type statistical models. As indicated earlier, our model specifications allowed for these KI impacts to be moderated by two respondent characteristics, K (Knowledge about vaccination) and S (Susceptibility to social influences). These interaction terms capture potential heterogeneity across respondents (mothers) in terms of the impact of KIs.

In addition, the changes in utility due to the level of the key influencers' support for the recommended schedule were calculated via one-way sensitivity analysis. Essentially, we map out the conditional main effect (i.e., interaction terms set to nil) of each KI on the utility of a vaccination schedule. And finally, we characterized the aggregate effect of the value of social level consensus for the recommended childhood vaccination schedule by building the cumulative key influencer group effect one by one (i.e. repeatedly adding one KI at a time), starting with the partner. Thus, these statistics concerning the impact of key influencers will give us a two-fold perspective on social influences: first, at the individual KI level, controlling for all other KIs; second, at the aggregate/group level, by demonstrating how influential multiple KIs can be.

## 3. Results

### 3.1. Respondents

In total 604 pregnant women aged 18 years and older participated in the online panel survey concerning childhood vaccination decisions. The Dynata panel provided 404 respondents, and another 200 were recruited from the Bounty Panel. These respondents had a mean age of 31 years (SD = 4.8) and were on average 26 weeks pregnant (SD = 9.0) (Table 2). About 53% of the sample had at least a bachelor's degree. Approximately 96% of the respondents reported that they were vaccinated as a child, and 37% of the respondents mentioned that they had one or more children at the time of the survey. The mean score on the vaccine hesitancy scale, according to our scale calculation method which generated sample values ranging from -20 (no hesitancy) to +15 (high hesitancy), was -10.3 (SD = 7.1), with 6.1% having a score higher than the threshold score suggesting behaviour consistent with high vaccine hesitancy (i.e., membership in class 1—we return to this topic shortly). Almost 97%, 91% and 87% of the sample had a partner, a friend with children, and a usual GP respectively (Table 3). Persons that were mainly considered as sources for health information were the usual GP followed by the usual other health care professional, the usual pharmacist, and the friend with children (97.0, 94.7, 87.1 and 70.4%, respectively). It is clear from Table 3 that respondents have a wide range of perceptions concerning who is a health information source for themselves, which is suggestive that these potential sources could be differentially impactful on their choices.

### 3.2. DCE results

Seventy-eight percent of the respondents only opted for the recommended schedule, whereas less than three percent of the respondents only opted for the delayed schedule or only opted-out for vaccination. Essentially, 117 out of 604 respondents (19.4%) could be persuaded to trade between the alternatives. When the LCM-2 model considered both childhood vaccination attributes and key influencers, a very high model fit was reached (pseudo-$R^2$ = 0.825) (Table 4). (Note that a 3-class model was also estimated, but a comparison of BIC measures ultimately led to the selection of the 2-class model.) The average class probabilities within the sampled population were 10.9% and 89.1% for latent classes 1 and 2, respectively. The probability to belong to a specific class depended strongly on the WHO vaccine hesitancy scale value. That is, membership in the classes is determined by responses to the 10 measurement

**Table 2. Respondents' characteristics.**

|  | N = 604 | % |
|---|---|---|
| Mean age (SD) | 31.1 (4.8) | Range: 18–46 |
| How far along is pregnancy (mean weeks (SD)) | 25.6 (9.0) | Range: 3–39 |
| Highest education |  |  |
| Post grad degree | 100 | 16.6% |
| Bachelor degree | 218 | 36.1% |
| High school only | 52 | 8.6% |
| Annual household income < $51,999 | 109 | 18.0% |
| $52,000 - $103,999 | 174 | 28.8% |
| $104,000 - $207,999 | 211 | 34.9% |
| >$208,000 | 35 | 5.8% |
| Number with children <18yrs already | 221 | 36.6% |
| Country of birth (top 3) |  |  |
| Australia | 430 | 71.2% |
| United Kingdom | 24 | 4.0% |
| India | 19 | 3.1% |
| Country of parent's birth (top 3) |  |  |
| Australia | 346 | 57.3% |
| United Kingdom | 44 | 7.3% |
| India | 21 | 3.5% |
| Vaccinated as a child (yes) | 581 | 96.2% |
| WHO vaccine hesitancy* | -10.3 (7.1) | Range: -20,+15 |

Note:

* Unit weights for the WHO vaccine hesitancy scale items were used, which can lead to a maximum total vaccine hesitancy value range from -20 to +20; the higher the score, the higher the perceived vaccine hesitancy.

items shown in Table 4 (section labelled *Classification Scoring Function*: *WHO_SAGE Vaccine Hesitancy Scale*), which are aggregated/summed together using the signed unit weights (±1's) also shown in Table 4. Making these weights explicit serves two functions: 1) it makes explicit that some items have positive or negative valence; and 2) shows that these weights were not

**Table 3. Characteristics of the key influencers.**

|  | Do you have... | | If yes, would you consider them a health information source? | |
|---|---|---|---|---|
|  | N | % (of total) | N | % (of people with...) |
| *Partner* | 584 | 96.7 | 381 | 65.2 |
| *Mother* | 550 | 91.1 | 368 | 66.9 |
| *Father* | 498 | 82.5 | 228 | 45.8 |
| *Friend with children* | 547 | 90.6 | 385 | 70.4 |
| *Friend without children* | 518 | 85.8 | 183 | 35.3 |
| *Usual GP* | 525 | 86.9 | 509 | 97.0 |
| *Usual other health care professional (e.g. community nurse)* | 207 | 34.3 | 196 | 94.7 |
| *Usual pharmacist* | 225 | 37.3 | 196 | 87.1 |
| *School teacher* | 233 | 38.6 | 108 | 46.4 |

**Table 4. DCE results based on an aggregate MNL and a 2-class ordered latent class model.**

| | Aggregate MNL | | 2-Class Ordered Latent Class Model | | | |
|---|---|---|---|---|---|---|
| Log Likelihood | -1734 | | -1392 | | | |
| Pseudo-Rho-squared | 0.782 | | 0.825 | | | |
| # Parameters | 50 | | 101 | | | |
| Deviance | 3468 | | 2783 | | | |
| AIC | 3368 | | 2581 | | | |
| BIC | 3788 | | 3430 | | | |
| | | | Class #1: | | Class #2: | |
| | All respondents | | Vaccine hesitant respondents | | Vaccine accepting respondents | |
| Class probability | 100.0% | | 10.9% | | 89.1% | |
| *Utility Function $V_j$* | $\hat{\beta}$ | | $\hat{\beta}_1$ | | $\hat{\beta}_2$ | |
| ASC Recommended vaccine schedule | -1.4023 | | -2.419 | | 0 | |
| ASC Split vaccine schedule | -1.1237 | | 0.333 | | 0.0244 | |
| ASC No vaccination | 0 | | 0 | | 0.2061 | |
| *# Vaccines Delayed—None* | 0 | | 0 | | 0 | |
| 1 Vaccine | -0.0107 | | -0.2329 | | 0.0051 | |
| 2 Vaccines | -0.0135 | | 0.1428 | | -0.0937 | |
| 3 Vaccines | 0.3147 | | 0.7122 | | 0.4195 | |
| *# Vaccine Sets Split Over 12 months—None* | 0 | | 0 | | 0 | |
| 1 Vaccine Set | 0.0597 | | -0.5071 | | 0.1243 | |
| 2 Vaccine Set | 0.0555 | | -0.2472 | | 0.2064 | |
| 3 Vaccine Set | -0.0381 | | 0.162 | | 0.0161 | |
| 4 Vaccine Set | -0.1349 | | -0.6343 | * | -0.204 | |
| *# Injections Compared to Recommended Schedule* | | | | | | |
| Less than | 0.1788 | | 0.7955 | | 0.2117 | |
| Same as | 0 | | 0 | | 0 | |
| More than | -0.3184 | | -0.3744 | | -0.2775 | |
| *# Clinic Visits Compared to Recommended Schedule* | | | | | | |
| Less than | -0.1514 | | -0.6812 | | -0.2397 | |
| Same as | 0 | | 0 | | 0 | |
| More than | -0.1253 | | 0.5371 | | -0.3596 | |
| *X = occurrences of preventable disease per 1,000 children* ln($X$/1000) | -0.414 | | -0.5086 | | -0.5152 | * |
| *X = occurrences of side effects per 100 children* | | | | | | |
| Linear L = $X$/100 | 0.1596 | | -0.2705 | | 0.0824 | |
| Quadratic = $L^2$ | -0.6916 | | -0.3843 | | -0.34 | |
| *Childcare enrolment for unvaccinated children* | | | | | | |
| No restrictions | 0 | | 0 | | 0 | |
| No enrolment during outbreak | -0.1686 | | -0.3893 | * | -0.1235 | |
| Cannot enrol at all | -0.0872 | | 0.1129 | | -0.1968 | |
| *Family Assistance Payments* | | | | | | |
| Continue as normal | 0 | | 0 | | 0 | |
| Reduced payments | -0.1074 | | -0.2705 | | -0.0947 | |
| Payments eliminated | -0.0351 | | 0.036 | | -0.1548 | |
| *ln($q_{gj}$): perceived allocation of Key Influencer to each vaccination option: [0,1]* | | | | | | |
| | $\alpha_{gj}$ | | $\alpha_{gj}^1$ | | $\alpha_{gj}^2$ | |
| ln(Partner) | 0.2231 | *** | 0.3966 | | 0.0856 | |

(*Continued*)

**Table 4.** (Continued)

| | Aggregate MNL | | 2-Class Ordered Latent Class Model | | | |
|---|---|---|---|---|---|---|
| ln(Mother) | 0.0035 | | 3.0485 | *** | 0.1696 | ** |
| ln(Father) | 0.0489 | * | -3.2923 | *** | -0.0061 | |
| ln(Friend w/Children) | 0.1789 | *** | -0.8193 | ** | 0.1304 | |
| ln(Friend w/o Children) | -0.0037 | | 2.2406 | *** | 0.0984 | |
| ln(Usual GP) | -0.1199 | *** | -0.7051 | | -0.0766 | |
| ln(Other Health Care) | 0.0543 | ** | 3.1293 | *** | -0.0037 | |
| ln(Usual Pharmacist) | 0.0797 | *** | 4.9391 | *** | 0.0498 | |
| ln(School Teacher) | -0.339 | *** | -0.0782 | | -0.4417 | *** |
| ln(Clergy) | 0.1521 | *** | -8.5758 | *** | 0.4507 | * |
| K*ln(Partner) | 0.0473 | | 9.0458 | *** | 0.1765 | |
| K*ln(Mother) | 0.1343 | *** | -7.4276 | ** | -0.1377 | |
| K*ln(Father) | -0.0196 | | 24.7012 | *** | -0.0286 | |
| K*ln(Friend w/Children) | -0.2213 | *** | 9.0899 | *** | -0.1867 | |
| K*ln(Friend w/o Children) | 0.1527 | *** | -13.4261 | *** | 0.051 | |
| K*ln(Usual GP) | 0.3494 | *** | 1.3968 | | 0.2397 | |
| K*ln(Other Health Prof) | -0.0988 | ** | -8.6384 | *** | -0.0347 | |
| K*ln(Usual Pharmacist) | 0.0548 | | -23.1202 | *** | 0.159 | |
| K*ln(School Teacher) | 0.5442 | *** | -2.545 | | 0.6769 | *** |
| K*ln(Clergy) | -0.3082 | *** | 19.0084 | *** | -0.5739 | * |
| S*ln(Partner) | -0.0432 | *** | 1.0798 | * | -0.0603 | |
| S*ln(Mother) | 0.072 | *** | 6.0791 | *** | 0.0371 | |
| S*ln(Father) | -0.0758 | *** | -4.5436 | *** | -0.1267 | *** |
| S*ln(Friend w/Children) | 0.0235 | * | -0.6454 | | 0.1183 | *** |
| S*ln(Friend w/o Children) | -0.0519 | *** | 1.9554 | *** | -0.1017 | *** |
| S*ln(Usual GP) | 0.0816 | *** | -1.229 | ** | 0.0929 | ** |
| S*ln(Other Health Prof) | -0.1109 | *** | -5.5454 | *** | -0.1023 | * |
| S*ln(Usual Pharmacist) | 0.0352 | * | 2.6599 | *** | 0.0649 | |
| S*ln(School Teacher) | 0.0488 | ** | 8.4356 | *** | -0.0108 | |
| S*ln(Clergy) | -0.0123 | | -11.6259 | *** | 0.1035 | |

| *Classification Scoring Function*: WHO_SAGE Vaccine Hesitancy Scale (higher score ⇒ greater hesitancy) | | | Responses on Likert scale (code for scoring function):<br>1 = strongly disagree (-2),<br>2 = disagree (-1)<br>3 = neither agree nor disagree (0)<br>4 = agree (1)<br>5 = strongly agree (2) | | | |
|---|---|---|---|---|---|---|
| Childhood vaccines are important for my child's health | | | | -1 | | |
| Childhood vaccines are effective | | | | -1 | | |
| Having my child vaccinated is important for the health of others in my community | | | | -1 | | |
| All childhood vaccines offered by the government programme in my community are beneficial | | | | -1 | | |
| New vaccines carry more risks than older vaccines | | | | 1 | | |
| The information I receive about vaccines from the vaccine program is reliable and trustworthy | | | | -1 | | |
| Getting vaccines is a good way to protect my child/children from disease | | | | -1 | | |
| Generally I do what my doctor or health care provider recommends about vaccines for my child/children | | | | -1 | | |
| I am concerned about serious adverse effects of vaccines | | | | 1 | | |
| My child/children does or do not need vaccines for diseases that are not common anymore | | | | 1 | | |
| *Cutoffs* | | | | | | |
| Tau(1) | | | | 1.1304 | *** | |

*(Continued)*

**Table 4.** (Continued)

| | Aggregate MNL | | 2-Class Ordered Latent Class Model | |
|---|---|---|---|---|
| Tau(2) | | | — | |

Notes: K = Respondent knowledge level about vaccination; S = Respondent susceptibility to social influences; MNL = Multinomial logit; AIC = Akaike information criterion; BIC = Bayesian information criterion; ASC = Alternative specific constant; ln = natural logarithm; GP = General practitioner; Prof = Professional.

estimated from the data, but defined so as to allow direct and easy subsequent use of the classification model in any advisory tools that might be developed from our results. The items themselves were coded according to the coding scheme also included in the table under the heading "Responses on Likert scale (code for scoring function)". The resulting WHO vaccine hesitancy score (= sum(weights * item responses)) ranges from -20 to +20, though empirically ranged from -20 to +15 in the sample (see Table 3). A user can apply the scoring mechanism above to determine class membership for a randomly selected mother in the target Australian sub-population, and thence have preference estimates for that mother without having to conduct a DCE. To be more precise, if the respondent had a vaccine hesitancy score of [-20, 1.1304], she belonged to class #2, but if she had a vaccine hesitancy score of [1.1304, +20] she belonged to class #1 (see Table 4 and Fig 1). A threshold $\tau$-value of 1.1304 on the WHO vaccine hesitancy was identified by the model as the cut-off value between class #1 and class #2.

The estimated coefficients for each latent class showed that the impact of key influencers on the maternal choice for childhood vaccination was massive (in magnitude and significance) compared to the impact of childhood vaccination attributes (Table 4). To exemplify, in Table 4, 'Friends with children' has a strong impact on the respondent such that if the respondent believes the friend with children would recommend vaccination, she would be more likely to vaccinate; however, this influencer's impact would be largely mitigated if the respondent is deemed more knowledgeable about vaccination, but the impact would be increased if the respondent is deemed more susceptible to social influences. This result, coupled with the extremely high fit of the 2-class model (as noted above, $R^2$ = 0.825), suggests that for these respondents and this choice context, the key influencers were so important in making their choices in the DCE, that the decision for childhood vaccination can be safely interpreted as being purely socially driven.

## 3.3. Impact of social influences on childhood vaccination decision

The marginal impact on the respondents' utility of a childhood vaccination schedule differed between key influencers. Based on the DCE coefficients, the partner of the gravid woman had by far the largest impact on the maternal choice for childhood vaccination, followed by the gravid woman's mother, and thirdly, friends with children (Fig 2). This result was partly confirmed by the direct survey question regarding how important the view of a certain key influencer was (convergent validity), where the partner of the gravid woman was also mentioned to be the most important key influencer (Fig 3). Respondents mentioned that all key influencers would likely (80% or more) recommend the government recommended childhood vaccination schedule profile (Fig 4). Except for the partner, Fig 5 shows that the gravid woman's utility for government recommended childhood vaccination schedule profile changed a little bit due to the level of support for the government schedule she perceived from a certain key influencer. However, if there was a high consensus between key influencers (i.e. a high group support), the odds ratio impact for government recommended childhood vaccination schedule profile increased substantially (Fig 6) and showed the value of social consensus.

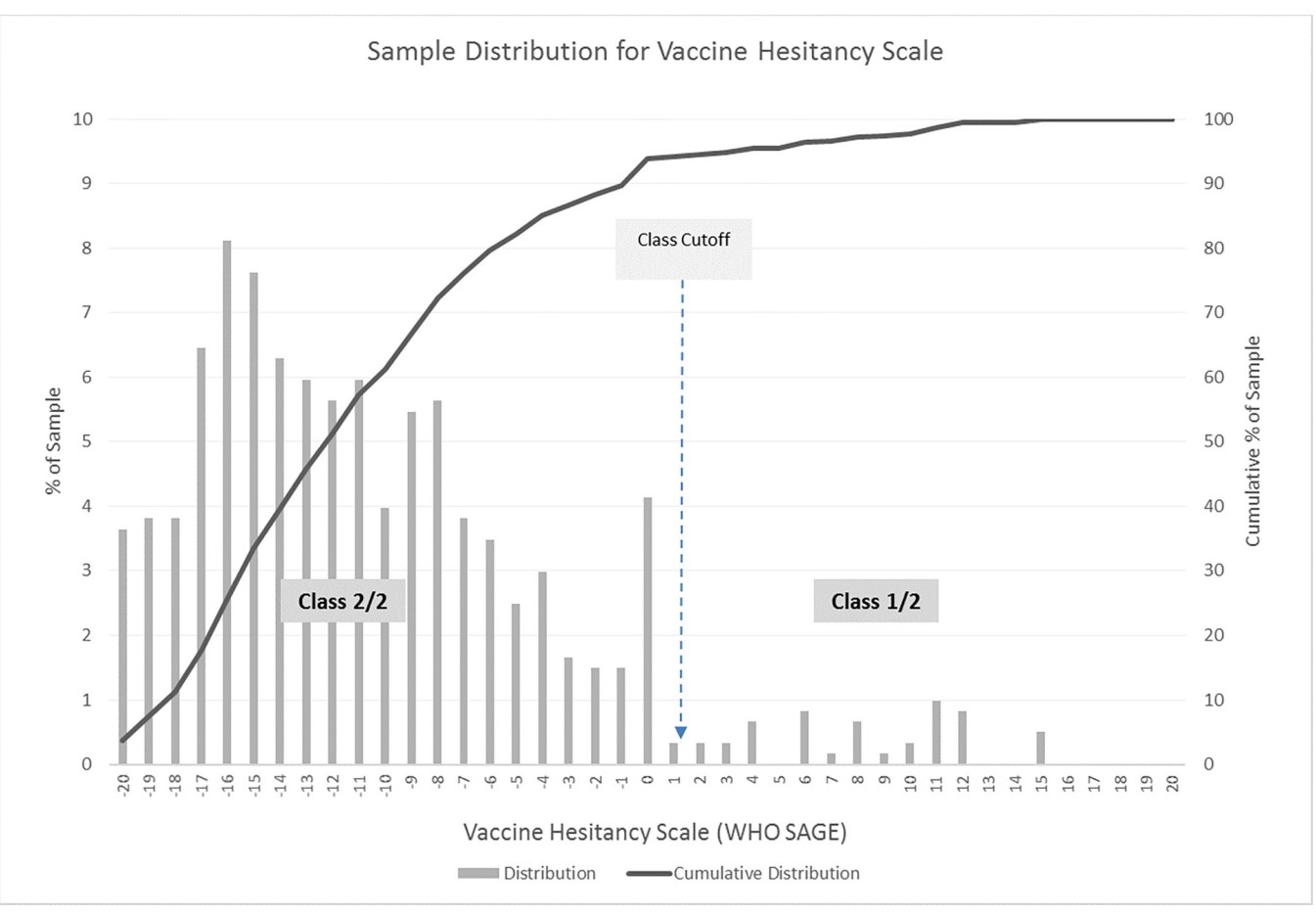

**Fig 1. Sample distribution for vaccine hesitancy scale.**

## 4. Discussion

Our study developed and empirically tested a choice model that accounted for social network influences in maternal choices for first year childhood vaccination in Australia. When the choice model took into account both childhood vaccination attributes and respondent beliefs about key influencer recommendations, our findings showed that the gravid woman's decision for childhood vaccination was essentially almost completely socially driven. The probability of opting for the government recommended childhood vaccination schedule profile depended strongly on the gravid woman's WHO vaccine hesitancy scale value, and on the level of social consensus between different key influencers. The marginal impact differed between key influencers, where the partner of the gravid woman had the largest impact on the maternal choice for childhood vaccination, followed by the gravid woman's mother and friends with children. Having knowledge about the decision-makers' social networks can change policy recommendations, since social networks generate significant influences on decision-makers. Namely, decision-makers exist in a matrix of social interdependence: health impact is not just on the individual but will potentially be felt throughout their social network.

In a qualitative study among MMR vaccination decisions, Poltorak et al. reported that actual choice outcomes depend not on a singular deliberative calculus and the information and education that informs it, but on contingent and unfolding personal and social circumstances in an evolving engagement [13]. This is completely in line with our quantitative results,

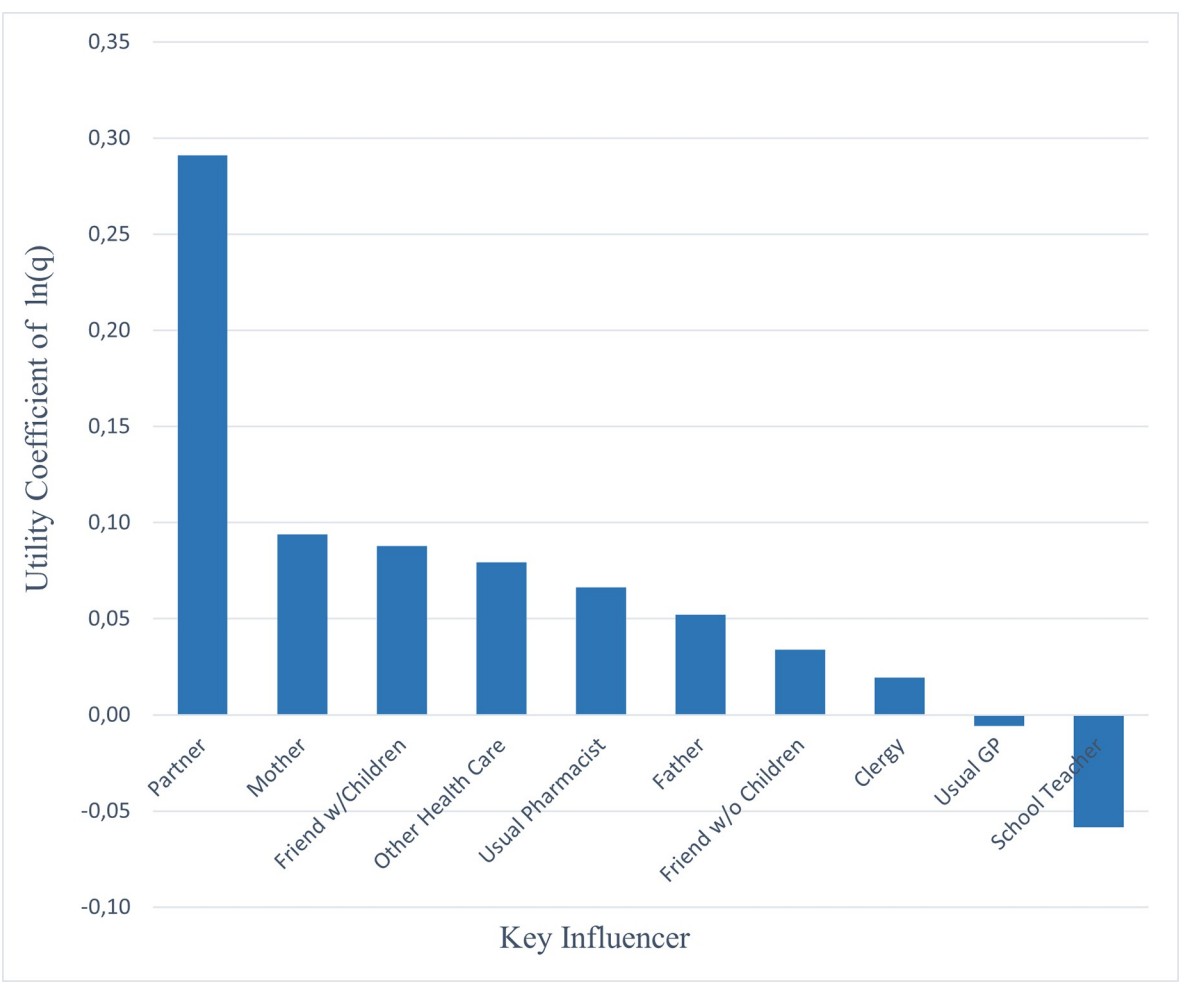

**Fig 2. Average marginal impact of a key influencer on utility of a vaccination schedule.**

which showed that the gravid woman's WHO vaccine hesitancy scale value matters in childhood vaccination choices and that these choices are not made in a social vacuum at all. Like Brunson [49], our study contributed to the evidence that the value of social network consensus should not be underestimated in childhood vaccination decision-making. And more forcefully, our results suggest the existence of healthcare decisions (this first-year childhood vaccination schedule among them) which may be wholly driven by social influences rather than service attributes. Hence, to increase the uptake of the childhood vaccination it is more effective to convince the social network than to reduce costs or clinic visits, for example. The influence of a policymaker via the social network will work very strongly if one can reach and convince the partner and mother of the gravid woman as these are the two persons who are most efficacious increasing the mother's likelihood to undertake childhood vaccination in Australia.

Given the authors' prior experience with DCEs in health, marketing, transport and environmental economics, the findings that the gravid woman's decision for childhood vaccination was essentially almost completely socially driven is an unusual result since theory would suggest that attributes are the strong drivers of choices. Instead, our DCE is demonstrating that social influences might be a strong explanator of behaviours in certain contexts. This observation should be strongly motivating for further research in social influences on individual

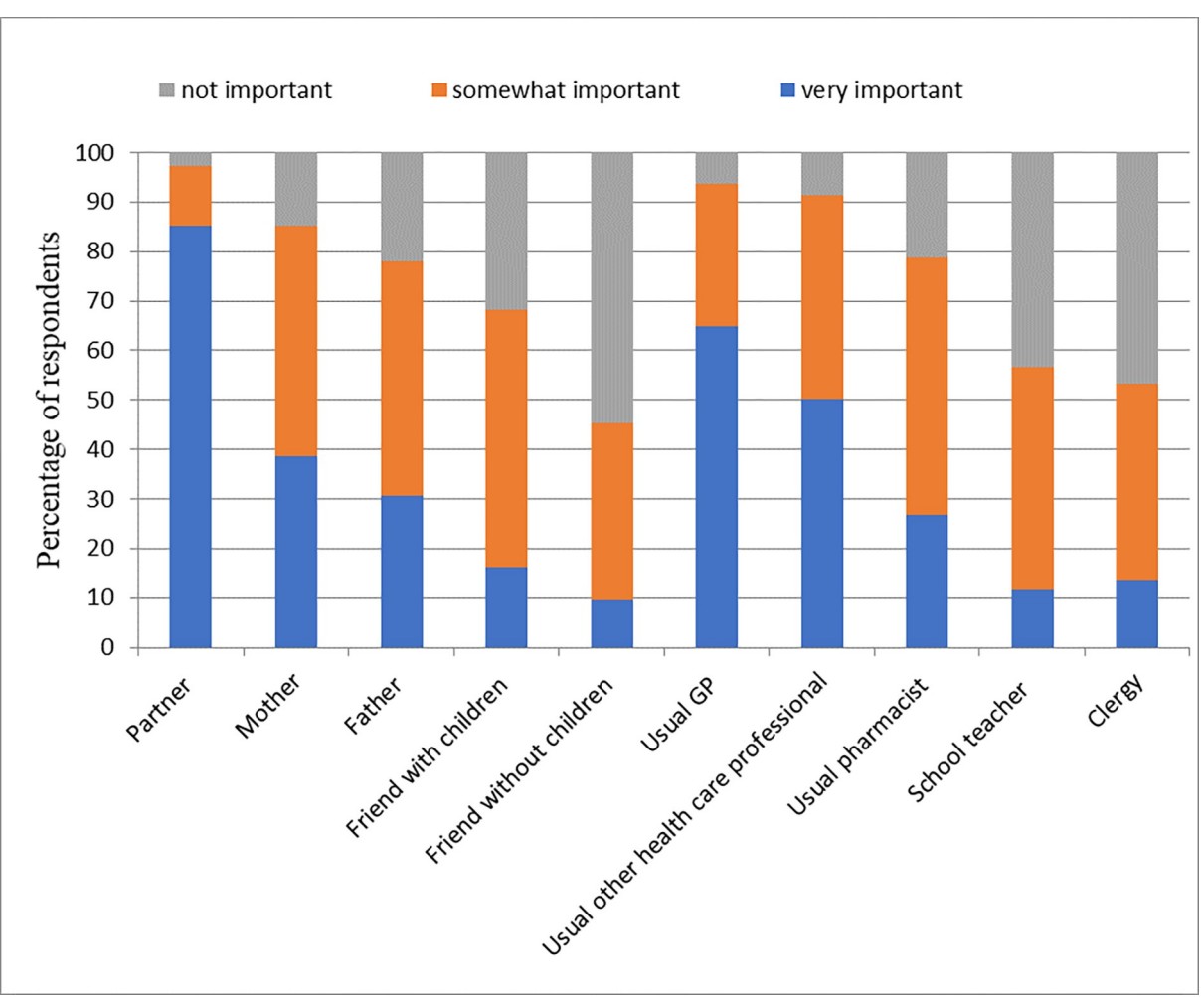

**Fig 3. Importance of key influencer views on respondent decision to vaccinate.**

choices, calling for further empirical work and choice paradigm evolution. That said, our study still showed that the DCE method is relevant here–even if the attributes of the vaccination were not the strongest drivers of stated choices for childhood vaccination in Australia— but also showed that DCE could benefit from extension. As shown in our study, this extension is needed to be able to jointly take health product/service/good attributes and social influences into account, while also considering the respondent's trade-offs between the variables, between key influencers and between variables and key influencers. Taking the social context in DCEs into account is especially relevant when the research question focuses on emotionally costly and/or complex decisions, or where there may be strong social norms. This includes–for example–a wide range of physician-patient interactions involving serious health conditions, multiple physicians deciding on treatment and other types of group decision making.

Our DCE outcomes and the outcomes of the direct survey question regarding how important the view of a certain key influencer was to the decision-maker showed a reasonable level of convergent validity. That is, on the one hand, it showed that both methods led to the conclusion that the partner of the gravid woman had the largest impact on the maternal choice for childhood vaccination. On the other hand, it showed that who were the second and third important key influencers depend on the method used.

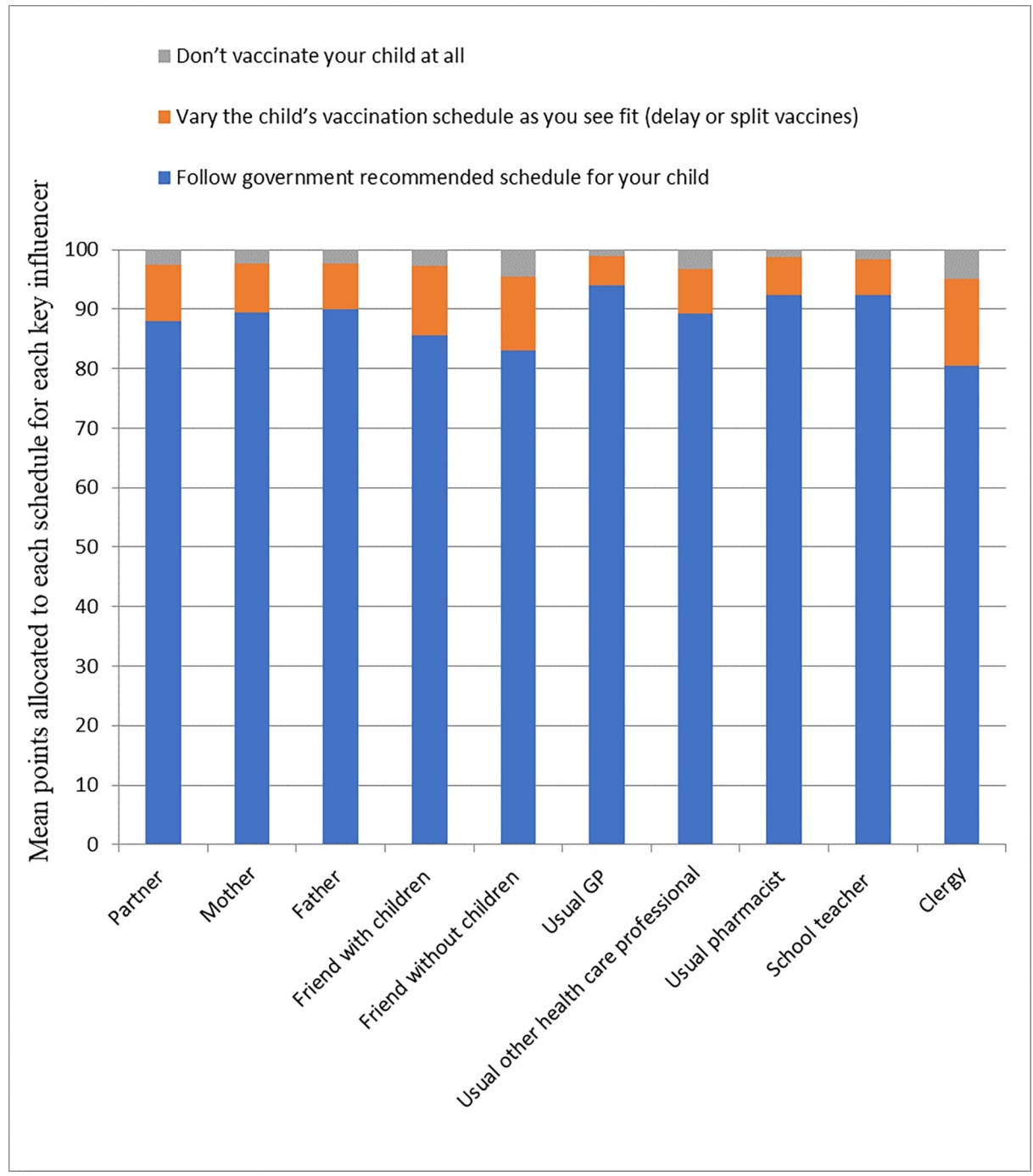

**Fig 4. How likely a respondent believes a certain key influencer would be to recommend each vaccination option.**

Although the choice model we developed in this study was able to account for social influences, we completely relied on key influencers preference information as perceived and reported by the gravid women only. Hensher et al. showed via their DCE study focusing on automobile purchase preferences, that sampling a single individual as a *representative* of the household's preferences is less appropriate than using preference information from the relevant group of decision makers in the household [50]. Although individual preference is easier

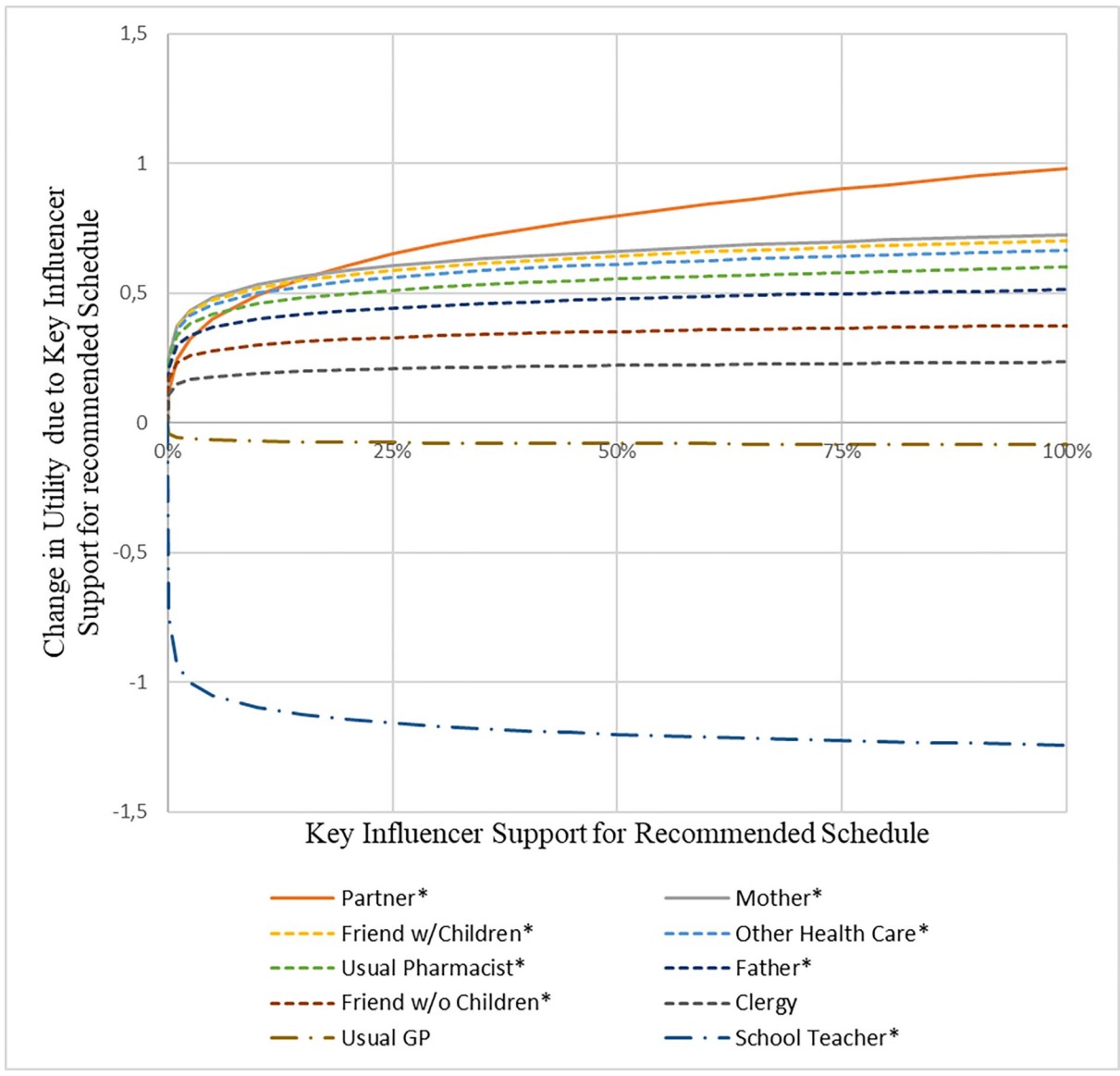

**Fig 5. Impact of key influencers on vaccination decision.**

to be measured and less expensive to obtain, we recommend the future exploration of whether the conclusion of Hensher et al. holds for childhood vaccination and/or other health care decisions too.

The current focus on the individual decision-maker in health economics and beyond has created a gap in choice theory to account for social influences. At best, there exist econometric corrections for biases that might arise from (somewhat amorphous) social influences, conceptualized in the form of an "influence field" impacting the decision maker [14]. Such efforts do not constitute a theory of choice. To be useful for scientific purposes, a theory of individual choice in the presence of influencers needs to formulate specific hypotheses about specific mechanisms for the influence on the decision-maker. This gap in choice theory development has the further consequence that econometric models fall short of capturing interpretable social impacts on the decision-maker, leading to difficulties in interpretation of health policy

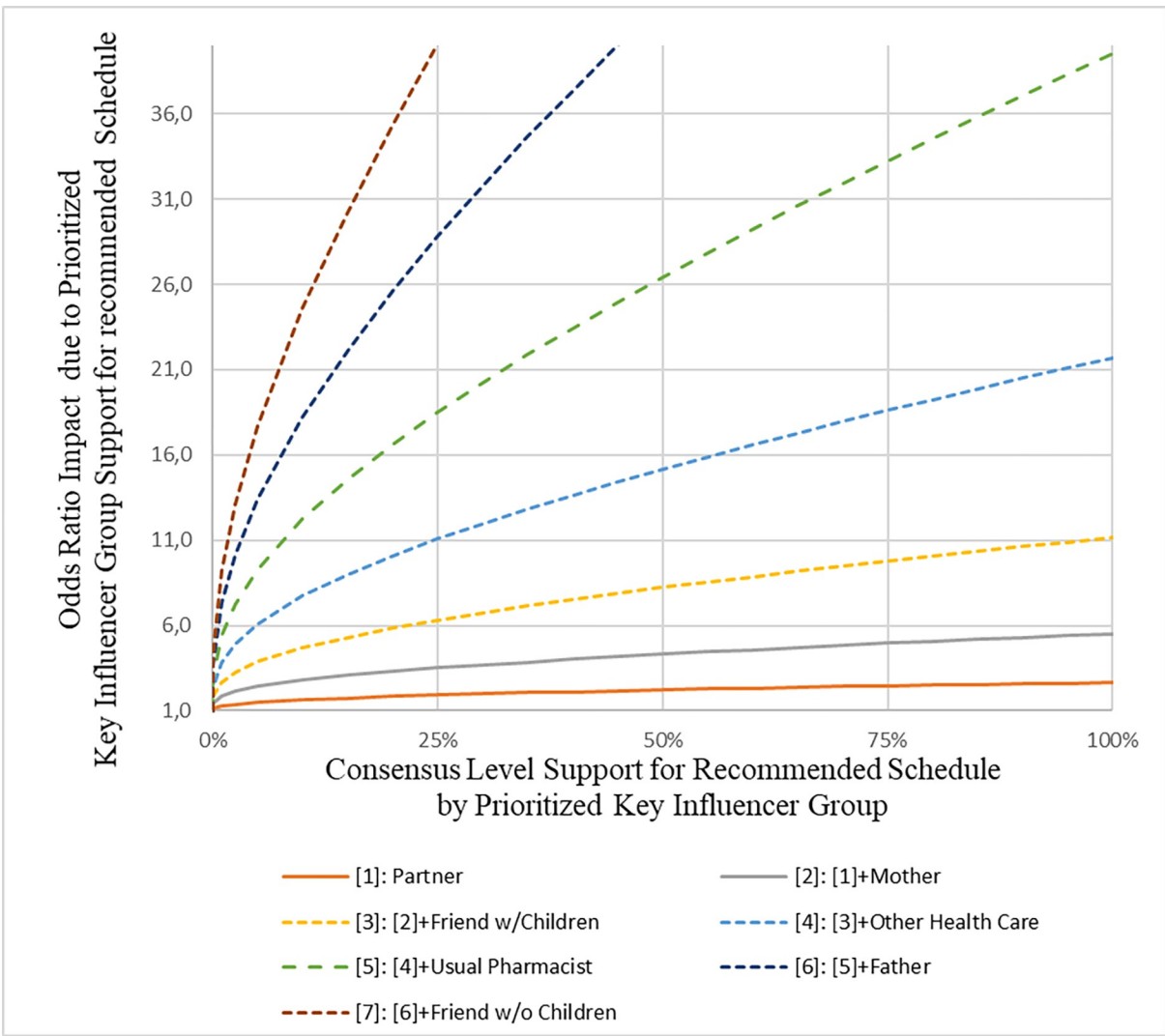

**Fig 6. The value of social consensus.**

implications. Finally, the lack of theory results in deficient and constrained measurement paradigms. Thus, despite the widespread recognition of social influences on individual choice behaviour in health care, and despite our and other choice modelers' attempts [8, 14] to account for social influences, an overreliance on the extant microeconomic framework for choice focused on a unitary decision maker has perhaps stymied choice theory evolution. Hence, moving towards a socially interdependent choice paradigm in a rigorous manner is crucial to support and reach accurate ex-ante evaluation of health care policies.

Our study has several limitations that may hamper the generalizability of the results. First, the study used online and specialized panels, and the results were based on one medical context only. Further research is therefore needed to determine whether our results hold outside online panels and in other medical contexts, where weaker or stronger underlying social norms may exist. Second, due to practical constraints we included only women who were pregnant at the time of participating in the survey. Expanding data collection to include partners of pregnant women and collecting and analysing the data with and without consideration

of joint decision-making would be ideal. Third, although a sample size of N = 604 is higher than an average DCE in healthcare [51] and is higher than Orme's rule of thumb suggests [52], we could not avoid the mandatory aggregation of certain levels to achieve parameter identification and to estimate robust parameters. It might be that this study–like almost all quantitative studies—could have benefitted from a larger sample size to give more detailed outcomes. Finally, the study focused on the childhood vaccination for the first 12 months of a child's life. As the role and influence of key influencers can change over time and with the age of the child, and parental motivators can change over time, assuming fixed individual's childhood vaccination preferences over many years might not be realistic.

In conclusion, our DCE study showed that the maternal decision for childhood vaccination was essentially completely socially driven, suggesting that the potential impact of social network influences can and should be considered in health-related DCEs, especially if there are likely to be strong underlying social norms dictating decision maker behaviour.

## Supporting information

**S1 Appendix. Discrete choice experiment design.**
(DOCX)

**S2 Appendix. Sample size calculation.**
(DOCX)

**S3 Appendix. Survey and sample.**
(DOCX)

**S4 Appendix. Integrating multiple influencers into an ordered latent class choice model.**
(DOCX)

**S5 Appendix. Discrete choice experiment analysis.**
(DOCX)

## Acknowledgments

We thank all respondents for their great contributions.

## Author Contributions

**Conceptualization:** Esther W. de Bekker-Grob, Kirsten Howard, Joffre Swait.

**Data curation:** Kirsten Howard, Joffre Swait.

**Formal analysis:** Joffre Swait.

**Funding acquisition:** Kirsten Howard, Joffre Swait.

**Methodology:** Esther W. de Bekker-Grob, Kirsten Howard, Joffre Swait.

**Project administration:** Esther W. de Bekker-Grob, Kirsten Howard.

**Supervision:** Esther W. de Bekker-Grob, Kirsten Howard.

**Visualization:** Esther W. de Bekker-Grob, Joffre Swait.

**Writing – original draft:** Esther W. de Bekker-Grob.

**Writing – review & editing:** Kirsten Howard, Joffre Swait.

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
