## [Decision Letter · Decision Letter 0]

12 Apr 2022

PONE-D-21-40314Identifying the impact of social influences in health-related discrete choice experimentsPLOS ONE

Dear Dr. de Bekker-Grob,

Thank you for submitting your manuscript to PLOS ONE. After careful consideration, we feel that it has merit but does not fully meet PLOS ONE’s publication criteria as it currently stands. Therefore, we invite you to submit a revised version of the manuscript that addresses the points raised during the review process.

In particular, one of the reviewers had serious concerns about how the experimental design was conducted and how the results were reported (reviewer 3).  As concerns the experimental design, the you should provide more information about the prior-setting procedure (i.e expert judgement process) and to what extent the assumed priors reflect the actual estimates.

Moreover, you should clarify the important inconsistencies that exist between the stated number of attributes/levels (2^3 x 4^1 x 5^1 x 10^1 x 16^1 x 18^1), what you  declared to consider as attributes/levels (i.e  "the number of injections over 12 months (levels: 0, 1,...,19)" have 20 levels;  visits to the clinic for vaccinations over 12 months (levels: 0, 1,...,16) have 17 levels; the chance the child will get a vaccine preventable disease (levels: 1/1000, 2/1000,…,12/1000) have 12 levels); and the way you included in the model: some attributes are considered as linear, (with quadratic term); an attribute had the ln transformation. I see several sources of unexplained arbitrariness in the model estimate. For instance, you declared that  the *18  parameters  of "number of injections  attribute" have been reduced to two for model identification*" and the same has been done for # Clinic Visits Compared to Recommended Schedule. Authors should clarify If the DCE has been performed (and the related priors) by considering the parameter number of injections  as "linear / continuous attribute " such as occurrences of side effects per 100 children, as 19 level attribute or 3 levels attribute. 

 The choice of an "ordered model" with two cut-off (3 levels) is unclear. Authors stated that "In each choice task, respondents were asked to opt for the alternative that appealed most to them. " Ordered model can be used in rated based tasks, or in ranking, while authors indicated in the text a choice-task.  

To conclude, all the mentioned issues should be solved to reach a publishable form, and thus there is no assurance of eventual acceptance. Indeed, statistical analyses need to be performed to a high technical standard and this is one of the main criteria for publication in PLOS ONE  https://journals.plos.org/plosone/s/criteria-for-publication#loc-3 .  

We look forward to receiving your revised manuscript.

Kind regards,

Prof. Francesco Caracciolo

Academic Editor

PLOS ONE

Journal Requirements:

“Grant support was from Australian Research Council Discovery Project DP140103966 to the third author, and from The Netherlands Organisation for Scientific Research (NWO-Talent-Scheme-Vidi-Grant No. 09150171910002) to the first author. The funders had no role in the study design, the collection, analysis and interpretation of data, in the writing of the manuscript or in the decision to submit the manuscript for publication.”

“Grant support was from Australian Research Council (https://www.arc.gov.au/) Discovery Project DP140103966 to JS, and from The Netherlands Organisation for Scientific Research (https://www.nwo.nl/en) (NWO-Talent-Scheme-Vidi-Grant No. 09150171910002) to EBG. The funders had no role in the study design, the collection, analysis, decision to publish, or preparation of the manuscript.”

Reviewers' comments:

Reviewer's Responses to Questions

**Comments to the Author**

1. Is the manuscript technically sound, and do the data support the conclusions?

Reviewer #1: Yes

Reviewer #2: Yes

Reviewer #3: Partly

2. Has the statistical analysis been performed appropriately and rigorously? 

Reviewer #1: Yes

Reviewer #2: Yes

Reviewer #3: Yes

3. Have the authors made all data underlying the findings in their manuscript fully available?

Reviewer #1: Yes

Reviewer #2: Yes

Reviewer #3: No

4. Is the manuscript presented in an intelligible fashion and written in standard English?

Reviewer #1: Yes

Reviewer #2: Yes

Reviewer #3: Yes

5. Review Comments to the Author

Reviewer #1: This manuscript reports a discrete choice experiment (DCE), designed to investigate the impacts of vaccine schedule characteristics and social network influences on decisions made by pregnant women regarding childhood vaccination. The results indicate that these decisions are almost entirely socially driven, providing insights that are both vital and timely as we continue to navigate our way through a global pandemic.

This is an exceptional study, with important implications for public health policy, DCE methodology, and the recognition of social influences on individual choice behaviour within health economics. I have few comments, most of which are aimed at making the manuscript more accessible to readers who are unfamiliar with the more technical aspects of DCE methodology. This is with a view to attracting as large an audience as possible for this very important work.

Specific points

p6, lines 124-136: It might be clearer to present these as bullet-points or in a table.

p6, line 146: Please insert “(MNL)” after “multinomial logit” – this is because the abbreviation MNL appears on p11 (line 213) without having been defined previously.

p10, lines 179-184: It would be useful to know a little more about the aims and basic designs of the Zingg et al, Larson et al and Holt et al studies, to explain why these were used as the sources for the questions about vaccines, vaccinations and susceptibility to social influences.

p11, lines 214-218: I think it would aid understanding for readers who are unfamiliar with latent class models (LCMs) if the underlying latent ordinal driver (the WHO vaccine hesitancy scale) were identified at this stage. In fact, the section that appears on p13 lines 257-262 could usefully appear here, as it provides a good explanation of why using a LCM is appropriate in this context.

p11-p14, Section 2.4 DCE Analysis: I wonder whether this section may be a little challenging for readers who are unfamiliar with DCEs; although I appreciate that this is difficult to address, given that DCEs are inherently quite technical and specialist. In particular, the text that appears on p13, lines 250-256, is critical to understanding how data from the questions on key influencers is incorporated into the model. Is there a way that this could be simplified for non-specialists?

p12, line 241: What does ASC stand for?

p22, lines 382-388: Would this fit better in the Discussion section?

p25, lines 460-465: I can’t follow the point that is being made here.

Finally, I thoroughly enjoyed reading this excellent manuscript. Thank you to the editors for assigning it to me. I’m looking forward to seeing this in print.

Reviewer #2: A well-written paper reporting a well-conceived study with a very competently performed data analysis. I think it is a significant contribution. I suggest some minor changes:

1) the natural log is sometimes reported as "ln", others as "LN", it should be consistent

2) lines 353 and 372, surely it is not R-squared, but pseudo-R-squared

3) In table 4, not sure what the scoring means and why it is reported

4) line 444 "to get uptake of the childhood vaccination", can this be written better?

5) lines 449-451, not clear what is intended here

6) importantly, some explanation should be provided about the model defined as "two-class ordered latent class model", I understand the panel, and I can see the MNL component, but it is unclear to me what the ordering is about, and how to interpret tau. What is the nexus between the (panel) probability of selections of the three vaccination options and the class membership, which appears to be linked to the ordering?

Reviewer #3: This is a very interesting paper which explores the effect of social influence on preferences for childhood vaccination among pregnant mothers in Australia. The paper concludes that social influences are the primary driver of decision making beyond any features of the vaccine schedule and delivery. Exploring other factors which drive decision making beyond service features is critical for expanding the predictive ability of choice experiments for policy makers, particularly for policies such as vaccination where social norms play a large role in decision making. Identifying approaches to incorporate social influence econometrically is important.

That being said, the paper has a few fundamental flaws:

Firstly, the DCE methodology seems potentially flawed, primarily because of the large number of attribute levels and unclear sample size estimation, given the authors extensive experience in DCE methods and design, the authors should clarify in the limitations some of the issues related to the design and sample size and how this may have influenced the results of the DCE findings. It may well be that decision making is entirely socially driven, however if there were not sufficient participants or if there were cognitive difficulties in answering the DCE it would not be prudent to draw such strong conclusions. Therefore, the survey/DCE tool should be included in the supplementary materials, details of poor responses and internal validity checks should be detailed. The sample size is important, particularly because so many results are “not significant”.

Second, the presentation of the methods and results sections in particular is very unclear. The methods, results and discussion are intermixed and therefore the reader must look for methods in the results section at times or find discussion points in the methods. The main results (Table 4) are not clearly presented and therefore make the results hard to interpret. The actual DCE findings are not discussed in any detail, presumably because little was “statistically significant”. Some methods sections are too long and should be presented more succinctly in the main paper with remaining econometrics included in supplementary materials. The authors and co-authors should review and make the methods and results sections as clear and succinct and well labelled as possible for a general readership to easily read and interpret. Given that this is not a health economics journal the authors should aim to make the paper readable for a general audience.

Detailed comments follow:

Methods:

The authors state this is a mixed methods study, there does not appear to be a qualitative component to the study, literature review is a common component of DCE surveys. Suggest the authors drop the text regarding mixed methods (which have their own methodology) and restrict to a DCE survey, which is what appears to have been done.

Rename the section 2.1 to “literature review and attribute selection”

Insert a table with final attributes and levels and reduce the text (lines 124-136 on page 6). Although a choice scenario example is presented in Table 1, it is not as easy to understand the design from visualizing only one choice task – both can be included in the paper or the current table 1 can go in supplementary materials.

Section 2.2 – DCE appears to have 8 attributes in the text but 7 in the scenario. Also, section 2.1 suggests an attribute with 19 levels but section 2.2 shows 18 levels. Authors should review the manuscript for consistency of methods and results.

The number of levels for several attributes seems excessive and brings into question the overall design of the experiment. Attributes with 12, 16, 19 levels are unusual for DCE’s because of a) the inability to obtain balance in the design and b) the inability of the participants to choose between the levels that are very similar for example 17 versus 18 visits to for vaccination. Even with the use of Ngene and development of an efficient design and blocking, this methodology is questionable given that each respondent answered 12 choice tasks and sample size was relatively small.. It appears that later these attribute levels were collapsed for this very reason. Please include some details in the limitations section regarding this potential design flaw and its implications for the results.

It appears that a labelled design was used, it is worth naming the DCE as such in the methods.

Could the authors please expand on the sample size estimation in the methods; although as noted, this is complicated for DCE’s – given the large design and the use of priors to develop the design, these same priors could have been used to estimate a sample size. Suggest the authors give more detail on the ideal sample size based on the design and priors/final estimates, even though affordability was a driver of the final sample size estimation. This will help to identify to what extend the analysis was limited by the samples size.

Was any piloting or cognitive interviews conducted, please report on these tool development stages.

Was the survey incentivized, assume so – this should be mentioned in the methods for the survey panel data.

Please include the original survey tool in supplementary materials.

How was internal validity determined – were any measures built into the design or explored afterwards, if so please describe - online surveys are notorious for poor responses.

The key influencer moderating characteristics scales and WHO vaccine hesitancy scale play a large role in the analysis and conclusions. Could the authors include the original scales and how they were analysed (scores generated from the original tool etc) prior to inclusion in the DCE modelling - in the supplementary materials.

Suggest section 2.4 be shortened and simplified for a general Plos One audience and that the detailed econometrics be included in the supplementary material to shorten this section and clarify the methods succinctly in the main paper.

It looks like the results are referenced in the methods eg. Table 4 in line 297 on page 11.

Please name the “customer developed software” used for analysis.

It appears that the methods for generating a WHO vaccine hesitancy scale cut-off are presented in the results. Please ensure that the methods for generating scale cut-offs are detailed in the methods sections (without presenting results in the methods section). It appears from the results that the WHO scale cut-off was developed based on the latent class membership – this should be detailed in the methods.

Results

3.1 Please include details on poor; incomplete responses and internal validity checks/random responses

Please add footnote to table 2 regarding how to interpret WHO vaccine hesitancy

On page 20 line 362 – 366 the authors present results related to adherence and uptake of vaccination schedules – it is unclear how these data can speak to adherence or uptake rates. Please remove this section from the results or rephrase.

Table 4

- It is unclear what all the results in the table represent – some sections have percentages other have utilities

– it is unclear what is being presented in the “Classification Scoring

Function: WHO_SAGE Vaccine Hesitancy Scale “ Please can the authors clarify this section of the table it is unclear what the reader is looking at.

Consider breaking this table up into multiple tables with clear column and row headings and appropriate footnotes. Or organize the table so that the reader can quickly distill what results are being evaluated and how to interpret them.

Authors should consider naming latent class 1 and latent class 2; this will help the reader to quickly understand which group is being referred to in the results throughout eg. “vaccine hesitant” vs “vaccine accepting” or something along these lines.

Consider placing the “Perceived allocation of Key Influencer to the vaccination option: [0,100]” section in a separate table and clearly label what exactly is being reviewed in this section.

Tables should be interpretable on their own, without having to comb through the methods and results to aid interpretation.

Authors should present either linear or quadratic results in the table based on the best fit to the data, this should be specified consistently for all attributes modelled continuously.

Please include confidence intervals for utilities. If space constraints an issue, reduce decimal places

The authors highlight that it is surprising that the attribute features had little influence on overall decision making and that the main decision drivers were related to social influences. Given that it is uncertain whether the sample size was sufficient to establish main utilities or latent classes, suggest the authors first explore the potential limitations of the sample size before making such claims. In addition, the authors identify early in the results that 19.4% of participants were willing to trade between alternatives, suggesting that some features of the scenarios must have influenced their decision making; could the authors reflect on this in the paper.

This section below starting on line 384 on page 22 in the results, should be in the discussion, not the results section:

“Given the authors’ prior experience with DCEs in health, marketing, transport and environmental economics, this is an unusual result since theory would

suggest that attributes are the strong drivers of choices. Instead, this particular DCE is

demonstrating that social influences might be a strong explanator of behaviours in certain

contexts. This observation should be strongly motivating for further research in social

influences on individual choices, calling for further empirical work and choice paradigm

evolution.”

Can the authors postulate potential reasons for some inconsistencies in the influence of partners. Table 4 shows partners playing a large role in main preferences but not for the latent class groups or not in combination with vaccine knowledge. It appears that the effect of partners varies across the analyses. Given that this is one of the strongest conclusions of the paper it would be helpful for the authors to fully interpret all the data presented related to the influence of partners across the various analyses, not only those that were statistically significant.

Although the results of the DCE were not statistically significant, the authors should present some narrative about the findings for the individual attributes and their levels, even if only non-significant trends are seen in the data. Currently there is no discussion of the actual DCE results. It is also hard to interpret the results when the final DCE design is not presented, it would be good to see what levels were presented for each attribute. For example, it appears that the “chance of your child getting a vaccine preventable disease” was important to participants though potentially not statistically significant. Confidence intervals may also highlight if the non-significant utilities are more around sample size than as the authors conclude that only social influence drives decision making.

Discussion/conclusion

Please expand the limitations to include potential issues with the DCE design/sample size/ online preference elicitation method. Please temper the strengths of the conclusions considering the possibility of flaws in the DCE design eliciting preferences for vaccine features. For example, it is hard to imagine that the risk of a child developing vaccine preventable illnesses is not important at all in decision making, which seems to be what the discussion is saying. It is more likely that it important but that social influences might be a stronger driver, for example.

6. PLOS authors have the option to publish the peer review history of their article (what does this mean?). If published, this will include your full peer review and any attached files.

Reviewer #1: No

Reviewer #2: No

Reviewer #3: No

---

## [Author Response · Author response to Decision Letter 0]

5 Jul 2022

Please see file "Response to Reviewers"

---

## [Decision Letter · Decision Letter 1]

29 Aug 2022

PONE-D-21-40314R1Identifying the impact of social influences in health-related discrete choice experimentsPLOS ONE

Dear Dr. de Bekker-Grob,

Thank you for submitting your manuscript to PLOS ONE. After careful consideration, we feel that it has merit but does not fully meet PLOS ONE’s publication criteria as it currently stands. Therefore, we invite you to submit a revised version of the manuscript that addresses the points raised during the review process.

Indeed, reviewer 3 identified a list of minor but essential comments to be addressed.

We look forward to receiving your revised manuscript.

Kind regards,

Francesco Caracciolo

Academic Editor

PLOS ONE

Journal Requirements:

Reviewers' comments:

Reviewer's Responses to Questions

**Comments to the Author**

1. If the authors have adequately addressed your comments raised in a previous round of review and you feel that this manuscript is now acceptable for publication, you may indicate that here to bypass the “Comments to the Author” section, enter your conflict of interest statement in the “Confidential to Editor” section, and submit your "Accept" recommendation.

Reviewer #3: All comments have been addressed

2. Is the manuscript technically sound, and do the data support the conclusions?

Reviewer #3: Yes

3. Has the statistical analysis been performed appropriately and rigorously? 

Reviewer #3: Yes

4. Have the authors made all data underlying the findings in their manuscript fully available?

Reviewer #3: Yes

5. Is the manuscript presented in an intelligible fashion and written in standard English?

Reviewer #3: Yes

6. Review Comments to the Author

Reviewer #3: The authors have addressed the comments sufficiently. Further revisions are related to length and readability for a PLoS One audience and additional use of supplementary materials.

1. Please include sample size calculations as described in the reviewer responses to a further appendix in supplementary materials and reference in the main manuscript.

2. Please further shorten and summarize methods section and add further supplementary methods appendixes. The current methods are 2-3 times the length of any other section. Given that this is not an econometrics journal, succinct methods will be sufficient for the manuscript and numerous supplementary appendixes can be used for detailed information

3. Please shorten the manuscript throughout by avoiding excessively lengthy descriptions and perspectives and defensive statements - review the manuscript and improve brevity of statements. For example - regarding incentivization: it is clear that the respondents were incentivized by the online panel companies according to their incentivization scheme. It would be sufficient to state this in a short sentence.

E.g., Revise: "We did not directly incentivise the survey, but the panel

companies had panellist reward systems that we had no control over. So, respondents were

incentivized according to the standard reward scheme used by the two panel companies."

To read: "The respondents were incentivized according to the online panel companies incentive scheme". Please review manuscript and use the word count judiciously. Although PloS allows for high word counts, excess can compromise the readability of the manuscript.

7. PLOS authors have the option to publish the peer review history of their article (what does this mean?). If published, this will include your full peer review and any attached files.

Reviewer #3: No

---

## [Author Response · Author response to Decision Letter 1]

20 Sep 2022

Response to Reviewers

We thank reviewer 3 and the editor for the additional evaluations, positive words and the valuable comments made, which we used to improve our manuscript further. Please find a point-by-point response to each comment presented below.

Reviewer #3 

“The authors have addressed the comments sufficiently. Further revisions are related to length and readability for a PLoS One audience and additional use of supplementary materials.”

1. Please include sample size calculations as described in the reviewer responses to a further appendix in supplementary materials and reference in the main manuscript.”

Response: Done. See S2_Appendix

2. “Please further shorten and summarize methods section and add further supplementary methods appendixes. The current methods are 2-3 times the length of any other section. Given that this is not an econometrics journal, succinct methods will be sufficient for the manuscript and numerous supplementary appendixes can be used for detailed information.”

Response: More than 600 words are moved from the Methods section to Appendices now (see S1_Appendix, S3_Appendix, S4_Appendix and S5_Appendix). Consequently, the length of the Methods section has been reduced substantially to improve the readability for the general reader without losing details for the reader who wants to obtain detailed insights in the econometric approach used.

3. ”Please shorten the manuscript throughout by avoiding excessively lengthy descriptions and perspectives and defensive statements - review the manuscript and improve brevity of statements. For example - regarding incentivization: it is clear that the respondents were incentivized by the online panel companies according to their incentivization scheme. It would be sufficient to state this in a short sentence. E.g., Revise: "We did not directly incentivise the survey, but the panel

companies had panellist reward systems that we had no control over. So, respondents were

incentivized according to the standard reward scheme used by the two panel companies."

To read: "The respondents were incentivized according to the online panel companies incentive scheme". Please review manuscript and use the word count judiciously. Although PloS allows for high word counts, excess can compromise the readability of the manuscript.”

Response: We thank the reviewer for this comment. We revised the manuscript accordingly. See also our response to reviewer’s comment #2 to increase the readability of the manuscript.

---

## [Editor Report · Decision Letter 2]

29 Sep 2022

Identifying the impact of social influences in health-related discrete choice experiments

PONE-D-21-40314R2

Dear Dr. de Bekker-Grob,

We’re pleased to inform you that your manuscript has been judged scientifically suitable for publication and will be formally accepted for publication once it meets all outstanding technical requirements.

Kind regards,

Francesco Caracciolo

Academic Editor

PLOS ONE
---

## [Editor Report · Acceptance letter]

10 Oct 2022

PONE-D-21-40314R2 

Identifying the impact of social influences in health-related discrete choice experiments 

Dear Dr. de Bekker-Grob:

I'm pleased to inform you that your manuscript has been deemed suitable for publication in PLOS ONE. Congratulations! Your manuscript is now with our production department. 

Kind regards, 

on behalf of

Dr. Francesco Caracciolo 

Academic Editor

PLOS ONE